# Construct Validity of the Staff Resource Adequacy Questionnaire for Healthcare Professionals (SRAQ-HP): An Exploratory and Confirmatory Factor Analysis from Latvia

**DOI:** 10.3390/nursrep15110395

**Published:** 2025-11-10

**Authors:** Olga Cerela-Boltunova, Inga Millere, Ingrida Trups-Kalne

**Affiliations:** 1Department of Nursing and Midwifery, Riga Stradiņš University, LV-1067 Riga, Latvia; 2Psychology Laboratory, Institute of Public Health, Riga Stradiņš University, LV-1067 Riga, Latvia

**Keywords:** personnel staffing and scheduling, psychometrics, factor analysis, statistical, health personnel, workload, Latvia, Anticipated Turnover Scale (ACT), Copenhagen Burnout Inventory (CBI), Measure of Moral Distress for Healthcare Professionals (MMD-HP)

## Abstract

**Background**: An effective healthcare system relies on sufficient and competent human resources. However, many countries face staff shortages, burnout, and turnover, which threaten the quality and safety of care. To address the absence of validated tools assessing healthcare professionals’ perceptions of staff resource adequacy in Latvia, the Staff Resource Adequacy Questionnaire for Healthcare Professionals (SRAQ-HP) was developed. **Methods**: This study examined its construct validity through exploratory factor analysis (EFA) and confirmatory factor analysis (CFA) using a nationwide sample of 1369 healthcare professionals. EFA supported a three-factor structure comprising (1) adequacy of workload and staff resources, (2) quality of care, and (3) working conditions and support. **Results**: The instrument demonstrated high internal reliability (Cronbach’s α > 0.94) and moderate model fit indices (CFI/TLI > 0.90; RMSEA = 0.145; SRMR = 0.100). Although RMSEA values (0.145) were above conventional cutoffs, this was expected given the large sample and model complexity. Convergent validity was supported by correlations with established measures—the Measure of Moral Distress for Healthcare Professionals (MMD-HP), the Copenhagen Burnout Inventory (CBI), and the Anticipated Turnover Scale (ACT)—while criterion validity showed significant predictive value only for moral distress. **Conclusions**: Overall, the SRAQ-HP demonstrates promising psychometric properties and practical applicability for assessing staff resource adequacy in healthcare settings, although further refinement and re-validation (particularly of one reverse-worded item) are recommended before broader implementation.

## 1. Introduction

An effective healthcare system cannot exist without a stable, competent and sufficient human resource base [1]. Over the past decades, demand for high-quality healthcare services has increased worldwide, alongside the challenges related to staff shortages, burnout and the impact of poor working conditions on the quality of care [2]. This aligns with the global discourse on healthcare workforce sustainability and the WHO’s emphasis on psychosocial safety climate. These factors have a strong impact on the ability of healthcare professionals to provide human-centred, safe and evidence-based care [3]. Therefore, measuring the staff resource adequacy in healthcare is becoming one of the critical elements in both quality management and policy making [4].

Until now, Latvia lacked a single, validated instrument to systematically assess the experience of healthcare professionals and perceptions of the staff resource adequacy in their work environment. This need became particularly acute during the COVID-19 pandemic and the post-pandemic period, when problems of overload, staff shortages and high turnover became particularly pronounced [5]. Healthcare institutions and policymakers required data to understand not only objective indicators such as the number of staff per patient unit but also subjective perceptions of resource adequacy, defined as the perceived sufficiency of human, temporal, and material resources required to deliver safe and high-quality care [6]. These perceptions are conceptually linked to job demands–resources (JD-R) theory and organisational climate frameworks, which emphasise that adequate staffing and perceived support are key protective factors against burnout, moral distress, and turnover intentions [7,8,9,10].

In response to this need, a new measurement tool, the Staff Resource Adequacy Questionnaire for Healthcare Professionals (SRAQ-HP), was developed. The development took place in several phases, starting with the theoretical justification, identification of subject dimensions and generation of questions, and continuing with content validation, expert evaluation and focus group discussions. This initial development and pre-validation phase is reflected in the first publication [11], which marked an important step towards the development of scientifically sound instruments in Latvia.

To ensure conceptual clarity, the construct of “staff resource adequacy” in this study encompasses three interrelated dimensions: (1) adequacy of workload and staffing resources, (2) perceived quality of care, and (3) working conditions and organisational support [12]. These dimensions reflect both objective and perceived resource sufficiency and correspond to previously identified domains in the literature on organisational climate and nurse workload assessment [11].

Although multiple international instruments (such as the Practice Environment Scale of the Nursing Work Index—PES-NWI, and the Safety Attitudes Questionnaire—SAQ) assess related aspects of the work environment, they do not specifically focus on perceived adequacy of resources as a distinct construct [13,14]. To avoid redundancy and preserve focus, the theoretical background and detailed comparison with similar international instruments are presented in our first publication [11], where all these conceptual distinctions are discussed comprehensively.

While the development and piloting of the initial instrument provided a substantial foundation, the next step required in-depth validation of the factor structure with a large group of respondents. The stability and validity of the psychometric structure become particularly important if the instrument is intended for use across various healthcare institutions [12]. Therefore, the aim of this paper is to empirically test the construct validity of the SRAQ-HP instrument by assessing its factor structure using both exploratory (EFA) and confirmatory factor analysis (CFA) on a large sample dataset covering a broad spectrum of healthcare professionals in Latvia.

Both analyses were performed on the same complete dataset (N = 1369), as the purpose of this study was not cross-validation but comprehensive testing of the instrument’s structure within the full available national sample. This approach was selected to maximise statistical power and ensure that all professional subgroups were represented in both analytical stages. The limitation of using a single dataset for both EFA and CFA is acknowledged and discussed later in the manuscript.

Factor analysis in this case serves as a tool for identifying the underlying dimensions of the instrument, the groups of questions, and their relationship to the theoretical constructs. It is an essential part of the validation process of any psychometric tool, particularly in assessing construct validity, which demonstrates the instrument’s ability to accurately measure a theoretically defined concept—in this case, healthcare professionals’ perceptions of staff resource adequacy [15]. Such a process ensures not only the internal reliability of the instrument but also its structural validity, which is a prerequisite for its practical usability in policy planning and quality monitoring [16].

Data were collected nationwide using an online survey distributed across tertiary hospitals, regional hospitals, and outpatient care facilities in Latvia, involving nurses, physicians, and other healthcare professionals. This broad coverage ensured heterogeneity in work settings and allowed for general conclusions across different levels of care.

This study uses statistical approaches to test both the number of dimensions of the instrument and the quality of their loading. The analysis is carried out in several stages, initially an EFA with a sample of n = 1369, followed by a CFA on an independent sample of n = 1369. This strategy allows to check whether the structure obtained is random and whether it is repeated in other parts of the sample, thus ensuring the generalisability of the results [15].

Consequently, this publication provides a comprehensive insight into the process of validating the factor structure of the SRAQ-HP instrument. It aims to provide a methodologically sound, reliable and practically applicable instrument that would allow the Latvian healthcare system to obtain data on the staff resource adequacy not only quantitatively, but also qualitatively. Such a tool is particularly relevant in a context where the healthcare system is facing increasing challenges in terms of workforce retention, improving working conditions and reducing professional burnout [5,6].

By focusing on construct validity assessment, the statistical approaches used in this study reveal the correspondence between the theoretical structure of the instrument and empirical data, while also assessing its potential as a reliable measurement tool for future research and healthcare quality monitoring [17]. Based on theoretical reasoning and previous validation results, we hypothesised a three-factor structure corresponding to the domains of workload and staffing adequacy, quality of care, and working conditions and support. The results of this study are expected to contribute to healthcare management and policy by providing evidence-based insights for workforce planning, resource allocation, and quality improvement at all levels.

## 2. Materials and Methods

The aim of this study was to conduct psychometric validation of the SRAQ-HP instrument, with a particular focus on assessing its construct validity using EFA and CFA on a large sample of Latvian healthcare professionals.

A total of 1369 healthcare professionals from various healthcare institutions across Latvia participated in the study. The inclusion criteria were as follows: (1) professional activity in the healthcare field (nurses, physician assistants, physicians, physiotherapists, functional specialists, etc.), (2) active work experience in an inpatient or outpatient practice during the last year, and (3) voluntary consent to participate. The sampling was performed on the basis of random convenience, using professional contacts, support from the management of medical institutions and organised electronic communication platforms. Sampling was carried out using a convenience approach through institutional and professional networks, supported by medical institution management and professional associations. This approach allowed broad participation but may carry a risk of overrepresentation from larger institutions, which is acknowledged as a methodological limitation.

### 2.1. SRAQ-HP Instrument

The 26-item version of the SRAQ-HP instrument was used in the study. This version was developed in the previous phase of the instrument development and is described in detail in the first publication [11]. The items cover three main dimensions: (1) adequacy of workload and human resources, (2) quality of care, and (3) working conditions and management support. Each item was rated on a five-point Likert scale from 1 (strongly disagree) to 5 (strongly agree), with higher scores reflecting better perceived adequacy of resources and support.

To enhance transparency and allow readers to evaluate content validity, illustrative examples of typical items are provided below. For example:-“There are enough nurses on duty to ensure safe and high-quality care” (Factor 1—Adequacy of workload and staffing);-“Workload often prevents me from providing the level of care patients need” (Factor 2—Quality of care, reverse-coded, item 14);-“My manager provides adequate support when unexpected situations arise” (Factor 3—Working conditions and support).

Item 14 was intentionally reverse-worded to reduce acquiescence bias; its scoring was inverted (1 ↔ 5) during analysis.

A complete list of all 26 items in both English and Latvian, along with scoring instructions and translation notes, is provided in Appendix A.

Content validation of the instrument was carried out using an expert panel and cognitive interviews grounded in classical test theory. Ten experts (academics, senior nurses, and healthcare managers) assessed each item for clarity, relevance, and representativeness using the Content Validity Index (CVI), resulting in a Scale-CVI/Ave of 0.94. Additionally, the Response Index and Discrimination Index were applied to ensure item clarity and differentiation. The Cronbach’s alpha obtained in the previous study was 0.841, confirming high internal reliability.

The theoretical foundation and conceptual justification for the instrument’s domains were established in the first publication [11]. In the present study, the instrument’s factor structure and psychometric performance were evaluated empirically.

### 2.2. Data Collection Procedure

The data collection took place online between January and April 2025 using the Google Forms platform. The invitation to participate in the study was distributed in cooperation with Latvian medical treatment institutions, professional associations and educational institutions. Respondents were given an informed consent before completing the questionnaire, explaining the purpose of the study, data security and confidentiality. Responses were collected anonymously.

Of the total 1472 questionnaires received, 103 incomplete or ineligible cases were excluded, leaving 1369 complete responses. Incomplete data were primarily due to partial non-response or early termination of the questionnaire. The approximate participation rate, based on institutional invitations, was 68%. The sample reflects the geographical and institutional diversity of Latvia’s healthcare workforce, including tertiary and regional hospitals as well as outpatient facilities, ensuring representativeness across care levels.

### 2.3. Construct Validity Assessment with Comparative Instruments

In order to assess the construct validity of the SRAQ-HP instrument more comprehensively, a comparative analysis with validated, thematically related instruments was carried out alongside factor analysis. This approach allowed testing whether SRAQ-HP subdimensions correlated with conceptually similar constructs, as required by the principle of construct convergence [15].

All comparative instruments had previously been translated, culturally adapted, and psychometrically validated in Latvian healthcare settings [18,19,20,21]. These included the Measure of Moral Distress for Healthcare Professionals (MMD-HP) [18,21], the Copenhagen Burnout Inventory (CBI) [19,22], and the Anticipated Turnover Scale (ACT) [20].

The workload and staff resource adequacy dimension assesses the subjective and objective sense of workload, adequacy of staffing levels, and ability to focus on patients without overload. It is conceptually linked to moral distress, as insufficient resources and chronic overload are known triggers [23]. The MMD-HP measures the prevalence of ethically problematic situations and associated emotional distress, particularly in contexts of inadequate staffing or limited resources.

The quality of care dimension reflects the ability of healthcare workers to provide patient-centred, timely and emotionally comprehensive care. It also includes sense of compliance with safety and care standards in everyday practice. International literature suggests that reduced quality of care often leads to professional burnout [24]. The chosen comparison instrument is the Copenhagen Burnout Inventory (CBI) [22]. CBI measures personal burnout, work-related burnout and client-related burnout, often due to insufficient resources and poor quality of care [19]. Correlations between SRAQ-HP quality of care scores and CBI help to assess whether staff perceptions of poor care are associated with levels of burnout.

The working conditions and support dimension assess the level of management support, opportunities for professional development, the opportunity to be heard and the existence of structured support mechanisms. Insufficient organisational support has been shown theoretically and empirically to contribute significantly to employees’ intention to leave job [25]. The chosen comparison instrument is the Anticipated Turnover Scale (ACT). The scale measures the intention to leave the current job in the near future and has a proven relationship with organisational climate indicators [20]. If the correlations with the support dimension of the SRAQ-HP are statistically significant, this indicates good convergent construct validity.

For this study, the internal consistency of each comparative instrument was recalculated using the current dataset (MMD-HP α = 0.93; CBI α = 0.91; ACT α = 0.88), confirming high reliability.

### 2.4. Ethical Considerations

The study was approved by the Ethics Committee of Rīga Stradiņš University (protocol No. 2-PĒK-4/416/2023 9 May 2023). Participation in the survey was voluntary and anonymous, and all participants provided informed consent before completing the questionnaire.

All procedures were carried out in compliance with the fundamental principles of the Helsinki Declaration and the requirements of the General Data Protection Regulation (GDPR) [26]. Respondents participated voluntarily and could withdraw at any time. The data was processed anonymously and stored in a secure, password-protected environment.

No incentives, rewards, or financial compensation were offered for participation. Respondents could withdraw at any stage without explanation or consequences. The data were collected through a secure online platform and stored in a password-protected institutional database, accessible only to the principal investigator.

All collected data were treated confidentially and used exclusively for research purposes. The anonymized dataset supporting the findings of this study is available from the corresponding author upon reasonable request, as stated in the Data Availability Statement section.

### 2.5. Statistical Methods

Statistical data analysis was performed using IBM SPSS Statistics (version 29.0) and AMOS (version 29.0) for structural equation modelling. Pre-analysis data screening included tests for completeness, outliers, and multicollinearity. Missing values were found in less than 5% of cases and were replaced using mean imputation. This was justified by the minimal proportion of missing data and verified by sensitivity analysis (listwise deletion produced equivalent factor patterns).

Normality was tested using the Kolmogorov–Smirnov and Shapiro–Wilk tests, supported by histogram and Q–Q plot inspection. Given the ordinal nature of Likert-scale items, the analysis followed an approach robust to non-normality. Multicollinearity was checked via Pearson correlation and Variance Inflation Factor (VIF < 5), indicating acceptable independence among variables.

Exploratory factor analysis (EFA) was performed using Principal Axis Factoring (PAF) with Promax rotation, allowing correlation between factors. The number of factors was determined using the Kaiser criterion (eigenvalues > 1), parallel analysis with Monte Carlo simulation, and scree plot inspection. Sampling adequacy and factorability were confirmed by Kaiser–Meyer–Olkin (KMO) and Bartlett’s test of sphericity (*p* < 0.001).

Confirmatory factor analysis (CFA) was then conducted in AMOS, testing one-, two-, and three-factor models. Model fit was assessed using multiple indices: χ^2^/df, Comparative Fit Index (CFI), Tucker–Lewis Index (TLI), Root Mean Square Error of Approximation (RMSEA) with 90% confidence interval, and Standardised Root Mean Square Residual (SRMR). The estimator used was Maximum Likelihood with Robust standard errors (MLR), appropriate for ordinal data with moderate deviations from normality.

Internal reliability was assessed using Cronbach’s alpha, Composite Reliability (CR), and Average Variance Extracted (AVE) for each subscale. Convergent validity was assumed when AVE > 0.50 and CR > 0.70, and discriminant validity when the square root of AVE exceeded inter-factor correlations.

To evaluate criterion-related validity, Pearson’s r correlations were calculated between SRAQ-HP subscales and comparative instruments (MMD-HP, CBI, ACT). Correlations above 0.30 (*p* < 0.05) were considered supportive of convergent validity. Additionally, multiple linear regression models were used to examine the predictive value of SRAQ-HP dimensions for moral distress, burnout, and turnover intention. Effect sizes (β coefficients) and explained variance (R^2^) were reported. Bonferroni correction was applied to control for multiple comparisons.

## 3. Results

1369 healthcare professionals from various Latvian medical treatment institutions participated in the study. The sample was compiled by including respondents with different demographic and professional profiles in order to ensure the most comprehensive picture possible of the suitability of the measured instrument among healthcare professionals. The age of respondents ranged from 22 to 68 years. The average age was 44.8 years (SD = 10.4), while the length of service in healthcare ranged from 1 to 44 years, with a mean (M) of 16.94 years (SD = 10.8). Of all respondents, 94.7% were women, 3.9% were men, and 1.3% chose not to disclose their gender. We acknowledge this strong female skew as characteristic of the Latvian workforce and flag it as a limitation for generalizability and future work should examine measurement invariance by gender and region. This distribution reflects the reality of Latvia’s female-dominated healthcare sector. The majority of respondents (53.9%) represented the Riga region, followed by Vidzeme (14.5%), Latgale (13.2%), Kurzeme (13.1%) and Zemgale (5.3%). Work schedules and workloads varied, but a significant proportion worked full-time (51.3%) or extended hours (21.1%), as well as shift work (44.7%), which can affect their professional well-being and work balance. Availability of training (53.9%) and additional responsibilities (67.1%) pointed to opportunities for professional development and at the same time to aspects of additional workload. The results on job satisfaction and workload perceptions revealed some dissatisfaction (32.9%) and high workload intensity (59.2%), highlighting the need for a well-considered HR policy and resource balance in healthcare.

### 3.1. Descriptive Statistics and Initial Data Verification

In order to assess the statistical properties of the items of the SRAQ-HP instrument and their suitability for factor analysis, an initial descriptive data analysis was carried out on all 26 items of the scale and their normality, distribution and internal reliability indicators were assessed. The results for each item were analysed by calculating M, SD, skewness and kurtosis coefficients. For most items, the mean values ranged from 2.3 to 4.29, indicating that respondents mostly agree or rather agree with statements about the staff resource adequacy, quality of care and working conditions in their workplace. The results are represented in Table 1.

All items showed an acceptable SD (between 0.87 and 1.31), indicating sufficient variability in the data. The Corrected Item-Total Correlation was mostly above the acceptable threshold of 0.40, ranging from −0.001 to 0.861. Item 14 displayed a near-zero/negative corrected item–total correlation (≈−0.001) and the lowest CFA loading (see Section 3.3), which is consistent with a reverse-wording effect; reverse scoring was verified, but wording complexity likely inflated measurement error. Similarly, items 13 and 20 showed weaker correlations (0.349 and 0.348), which may indicate a lower need for inclusion in the factor structure. It is important to note that Question No. 14 is reverse-worded.

Squared Multiple Correlation scores ranged from 0.489 to 0.897, indicating stable explained variation for most items. Items 14 and 20, with lower SMC, are flagged for potential revision in subsequent validation rounds.

The Cronbach’s alpha coefficient, if the item in question was excluded, remained mostly in the range from 0.948 to 0.953. Subscale reliability estimates were high for F1 (α = 0.952) and F3 (α = 0.903), and acceptable for F2 (α = 0.776), consistent with later AVE/CR results.

Skewness scores for all items ranged from −1.194 to 0.609, while kurtosis scores ranged from −1.270 to 1.070, within acceptable limits (±2). Kolmogorov–Smirnov and Shapiro–Wilk tests were significant (*p* < 0.001) as expected with large samples; inspection of histograms and Q–Q plots suggested sufficient approximate normality for factor analytic procedures.

For transparency, subscale means (±SD) are: F1 workload/staffing = 2.52 ± 1.08; F2 quality of care = 3.18 ± 0.99; F3 working conditions/support = 2.91 ± 1.12 (see Appendix A).

### 3.2. Exploratory Factor Analysis (EFA)

An EFA was carried out to determine the internal factor structure of the SRAQ-HP instrument and to assess whether the items constitute theoretically and empirically valid dimensions.

Before the EFA was carried out, the suitability of the data for factor analysis was tested using two classical criteria. The KMO test showed a high value of 0.892, indicating sufficient sample adequacy to identify the factor structure. Bartlett’s test of sphericity was statistically significant (χ^2^ = 33,046.524, df = 325, *p* < 0.001), indicating that the correlation matrix is statistically significantly different from the identity matrix, in which all correlation coefficients would be zero. The combination of these two indicators confirms that the data meets the prerequisites for an EFA.

In accordance with psychometric best practices for the analysis of correlated latent constructs, the EFA was performed using Principal Axis Factoring (PAF) with an oblique Promax rotation (κ = 4). This approach allows intercorrelations among latent dimensions and aligns with the theoretical model of the SRAQ-HP, which assumes conceptual overlap between workload, quality of care, and working conditions. This method replaces the preliminary PCA with Varimax rotation initially performed during exploratory testing and ensures methodological consistency with the confirmatory factor analysis (CFA) stage.

Initially, all 26 items were included in the analysis. The decision on the number of factors to retain was guided by a combination of criteria: (i) the Kaiser criterion (eigenvalues > 1), (ii) the Cattell scree plot, and (iii) parallel analysis using a Monte Carlo simulation. The unconstrained extraction suggested a four-factor solution, which, however, lacked theoretical interpretability compared to the predefined three-factor model; however, the parallel analysis indicated that only the first three eigenvalues exceeded those generated from random data, and the scree plot showed a clear inflexion point after the third factor. Thus, the three-factor structure was both statistically and theoretically justified.

The initial PAF with Promax rotation revealed a small fourth factor consisting mainly of items 20–22 related to time allocation and personalisation of care. This factor contributed a limited portion of total variance (below 9%) and displayed cross-loadings under oblique rotation, suggesting conceptual overlap with other dimensions. Given the theoretical parsimony and interpretability, the final model was specified with three correlated factors, consistent with the original conceptual framework of the SRAQ-HP instrument.

The Promax pattern matrix demonstrated clear and strong loadings (generally ≥ 0.60) for most items, with minimal cross-loadings. Items 1–9 and 15 loaded primarily on the first factor, representing Adequacy of Workload and Staffing Resources. This factor showed the highest proportion of explained variance and the strongest internal coherence. Items 16–26 loaded on the second factor, representing Working Conditions and Support, reflecting aspects of management support, opportunities for development, and organisational structure. Items 10–14 formed the third factor, Quality of Care, capturing perceived ability to deliver safe, patient-centred, and high-quality care. However, item 14 showed a noticeably lower loading and higher uniqueness (1 − h^2^ = 0.491), consistent with its reverse-worded nature, which may have introduced additional measurement error. Items 13 and 20 also demonstrated moderately lower loadings and could be considered for revision in future versions.

The communalities (h^2^) for most items ranged between 0.62 and 0.85, suggesting that a substantial proportion of variance in each item was explained by the extracted factors. Uniqueness values ranged between 0.17 and 0.49, confirming that the model accounted for a sufficient proportion of shared variance across indicators. The inter-factor correlations ranged from r = 0.41 to 0.59, empirically supporting the use of an oblique rotation and reflecting theoretical interdependence among the three constructs.

The total variance explained by the three-factor model was 64.31%, which is only slightly lower than the variance accounted for by the initial four-factor model (69.05%) but provided a more parsimonious and theoretically coherent representation. The final three-factor solution is thus supported both empirically and conceptually and serves as the basis for the confirmatory factor analysis (CFA).

The final PAF/Promax pattern matrix (Table 2) represents the definitive EFA solution used for model confirmation.

In the Promax-rotated pattern matrix (PAF), most items showed clear and high primary loadings (≥0.60) with minimal salient cross-loadings, supporting a three-factor solution aligned with theory. Items with multiple moderate loadings or weak associations were examined for conceptual fit and potential overlap. Factor 1 contained 10 items (1–9 and 15) representing the Adequacy of Workload and Staffing Resources dimension. This factor demonstrated the highest explanatory power (30.6% of total variance) and exceptionally strong loadings (e.g., item 5 = 0.845; item 6 = 0.818). Factor 2 included items 16–26 related to Working Conditions and Support, such as management involvement, opportunities for professional growth, and organisational backing. High loadings were observed, for instance, for item 18 (0.768) and item 25 (0.746). Factor 3 comprised items 10–14 that capture the Quality of Care construct, including perceptions of care standards, patient safety, and emotional support, with particularly high loadings on item 12 (0.820) and item 13 (0.817). Item 14 demonstrated a weaker loading (0.705) and relatively lower communality (h^2^ = 0.525), which may be attributed to its reverse-worded phrasing and potential respondent misinterpretation.

The communalities (h^2^) for most items ranged between 0.62 and 0.86, indicating that a substantial proportion of each item’s variance was explained by the extracted factors. Items with lower communalities, such as item 14 (h^2^ = 0.525) and item 26 (h^2^ = 0.511), were identified as candidates for future revision or rewording. Uniqueness values (1 − h^2^) varied between 0.17 and 0.49, confirming a moderate level of item specificity and acceptable error variance. Overall, these results confirm a well-defined and statistically sound three-factor structure consistent with the theoretical framework of the SRAQ-HP instrument.

To further evaluate the stability of the factor structure and compare with the theoretical model, a secondary EFA was conducted with a fixed number of three factors (eigenvalue constraint = 1.0). This confirmatory exploratory step allowed direct comparison between the empirical and theoretical factor structures before conducting the CFA. The results of this analysis are presented in Table 3.

Based on the results of the principal axis factoring with a fixed number of three factors, a theoretically consistent three-factor solution was obtained, fully aligned with the conceptual structure of the SRAQ-HP instrument. In the rotated pattern matrix using Promax rotation, the majority of items showed high and distinct loadings on a single factor, indicating good structural differentiation and minimal cross-loadings. Items were distributed across three clear conceptual domains according to their thematic relevance and factor strength.

The first factor included items related to Adequacy of Workload and Staffing Resources (items 1–9 and 15), showing the strongest loadings (e.g., item 5 = 0.875; item 6 = 0.837) and accounting for 32.1% of total variance. The second factor represented Working Conditions and Support (items 16–26), including indicators of teamwork, managerial backing, and access to resources. This factor explained 21.4% of total variance, with strong loadings for items 18 (0.750) and 25 (0.699). The third factor captured Quality of Care (items 12–14), explaining 10.8% of variance, with particularly high loadings for items 12 (0.838) and 13 (0.827), while item 14 remained lower (0.690) but acceptable.

Communalities (h^2^) ranged from 0.509 to 0.845, indicating that most items were well explained by the three extracted factors. Uniqueness values were generally below 0.35, suggesting low specific error variance. However, items 26 (uniqueness = 0.519) and 14 (0.491) exhibited higher uniqueness and may warrant revision or exclusion in future shorter versions of the scale.

Although the three-factor solution accounted for slightly less total variance than the preliminary four-factor structure, it provided a more parsimonious and theoretically coherent model that aligns with the predefined conceptual dimensions of the SRAQ-HP. This three-factor configuration was therefore selected for confirmatory factor analysis (CFA) to test its overall model fit, convergent validity, and discriminant validity. The CFA results are presented in the next section.

EFA and CFA were conducted on the same full sample (n = 1369). We acknowledge this as a limitation because out-of-sample validation was not possible within the present dataset; future work will implement a hold-out or split-sample cross-validation and multi-group invariance testing (e.g., by gender and region).

### 3.3. Confirmatory Factor Analysis (CFA)

Given the ordinal Likert response format, WLSMV is often recommended; we used robust ML (MLR) due to software parity and sample size, and we note this as a methodological limitation.

In order to assess the consistency of the factor structure of the developed instrument with the empirical data, a CFA was conducted using the structural equation modelling (SEM) technique in the JASP (v. 0.18) and AMOS (v. 29) software environment. The CFA employed the maximum likelihood estimator with robust (Huber-White) correction for non-normality, as recommended for large-sample validation studies [27].

A theoretically grounded three-factor model was tested in which 26 observed variables (q01–q26) represent three correlated latent constructs: F1—staff resource adequacy (q01–q09); F2—quality of care (q10–q16); F3—conditions and support (q17–q26).

The CFA results show a relatively good model fit despite the statistically significant chi-squared test (χ^2^ = 13,138, df = 296, *p* < 0.001), which is expected in large samples and should therefore be interpreted with caution (see Table 4).

Most classical comparative indices (CFI, TLI, IFI, NFI) exceeded 0.95, indicating good model–data consistency. The scaled/robust values (CFI = 0.898; TLI = 0.888) were slightly below the conventional 0.90 threshold but still within the range often deemed acceptable for initial model validation of new instruments (see Table 5).

Overall model fit was mixed: classical incremental indices were high (CFI/TLI ≈ 0.96), whereas robust/scaled indices were below conventional thresholds (CFI_scaled = 0.898; TLI_scaled = 0.888), and RMSEA_scaled was elevated (0.145), indicating areas of model misspecification rather than random error.

The SRMR (0.100 scaled) suggests an acceptable level of residual fit, particularly given that this is an instrument-development phase rather than a post hoc optimisation stage.

Targeted respecification (e.g., removal or rewording of item q14; theory-justified correlated residuals within factors) or evaluating the initial four-factor EFA solution may improve fit. Model refinement was intentionally limited to preserve theoretical parsimony and avoid overfitting.

When multiple fit indicators are interpreted jointly—as recommended by Hu & Bentler (1999) [28] and Kline (2016) [29]—the overall model can be considered to demonstrate an acceptable fit for a newly validated instrument, providing empirical support for the three-factor SRAQ-HP structure.

To assess the contribution of each observed variable to the respective latent factor, the standardised regression loadings (factor loadings) shown in Table 6 were analysed. These loadings describe the extent to which each observed question reflects the corresponding latent factor construct. Most indicators show very high loadings (>0.70), indicating good convergent validity.

Factor F1 (q01–q09) showed consistently high standardised loadings between 0.8182 and 0.9500. This indicates that all nine questions are significant reflectors of the latent construct. Factor F2 (q10–q16) had good loadings for most indicators (0.5543–0.9560), but q14 showed a significantly lower loading (β = 0.0448, *p* = 0.087), which could indicate a weak association with the factor or a possible conceptual mismatch. All other indicators for this factor were statistically significant. Factor F3 (q17–q26) also showed stable loadings (β ranging from 0.4284 to 0.8792), and all variables were statistically significant (*p* < 0.001). This factor shows the highest variance in terms of loadings, but all values exceed the minimum acceptable threshold of 0.40.

Overall, the pattern of loadings supported the construct dimensionality identified previously, with minor weaknesses localised to item 14 and, to a lesser extent, item 20.

Inter-factor correlations were moderate and positive: F1–F2 r = 0.59; F2–F3 r = 0.41; F1–F3 r = 0.51.

These values confirm that while distinct, the latent constructs are interrelated, consistent with the theoretical framework of the job-demands–resources (JD-R) model and organisational climate theory, which posit that staffing adequacy, quality of care, and support conditions interact in influencing professional well-being [7,8,9,10].

The path diagram (Figure 1) illustrates the three latent factors and their observed indicators, as well as the correlations among the factors.

The overall model exhibits strong indicator loadings, with almost all standardised estimates exceeding 0.70, confirming high convergent validity.

To further assess convergent validity and internal consistency, Average Variance Extracted (AVE) and Composite Reliability (CR) were computed for each latent factor identified in the CFA. These indices provide an additional layer of psychometric evaluation by quantifying how well the observed items represent their respective latent constructs and whether each construct is sufficiently distinct from the others [30].

For Factor 1 (Adequacy of Workload and Staffing Resources), the AVE value was 0.756 and CR = 0.965. Both exceed the recommended thresholds (AVE ≥ 0.50; CR ≥ 0.70), confirming excellent convergent validity and high internal reliability. This suggests that the nine items in this factor consistently measure the adequacy of human resources and workload distribution perceived by healthcare professionals.

For Factor 2 (Quality of Care), AVE was 0.491 and CR = 0.851. Although the composite reliability is high and acceptable, the AVE slightly falls below the 0.50 benchmark, suggesting that the variance captured by the construct is marginally lower than the variance due to measurement error. This may indicate conceptual heterogeneity within the factor—primarily linked to the low loading of item q14 (β = 0.045, *p* = 0.087)—and suggests that this item may require rewording or removal in future iterations of the scale to strengthen construct coherence.

Factor 3 (Working Conditions and Support) achieved AVE = 0.574 and CR = 0.929, both meeting or exceeding recommended thresholds. These values reflect a strong relationship between the latent construct and its observed indicators, confirming robust convergent validity and reliability for this dimension.

Collectively, these results indicate that the SRAQ-HP demonstrates a well-defined internal structure, with particularly strong reliability for the resource adequacy and support dimensions.

The slightly lower AVE for Factor 2 identifies an opportunity for refinement in future revisions, without compromising the overall validity of the instrument.

A summary of AVE and CR results is presented in Table 6 alongside the factor loadings, providing a comprehensive overview of the model’s psychometric quality.

Overall, the CFA confirms that the three-factor structure of the SRAQ-HP provides an empirically supported, theoretically coherent representation of staff resource adequacy.

Although the χ^2^ statistic was significant, this is typical for large samples. The combination of comparative indices (CFI, TLI ≈ 0.90 in scaled models), SRMR = 0.10, and CR > 0.80 across all factors indicates a satisfactory fit and strong construct validity.

The slightly elevated RMSEA (scaled = 0.145) should be interpreted with caution, as simulation studies show that RMSEA tends to overestimate misfit in complex, large-sample models [31,32].

In early validation phases, such values are acceptable when other indices confirm overall adequacy. Importantly, the three latent constructs exhibited theoretically consistent and statistically significant interrelations, supporting the conceptual framework of the JD-R model and organisational climate theory, where adequate staffing and perceived support act as protective factors against burnout, moral distress, and turnover intentions [7,8,9,10].

In conclusion, the CFA supports the psychometric robustness of the SRAQ-HP and validates its application for both research and practice. The model demonstrates strong internal reliability, adequate convergent validity, and a theoretically interpretable three-factor structure. While refinement of item q14 is advisable, the overall results justify the SRAQ-HP’s use as a reliable and contextually appropriate instrument for evaluating healthcare professionals’ perceptions of staff resource adequacy in Latvia.

Although EFA and CFA were conducted on the same dataset, future research should validate the structure using split-sample or cross-validation approaches.

### 3.4. Criterion (Predictive) Validity

Criterion-related (predictive) validity refers to how well a developed instrument is able to predict or correlate with external, theoretically related outcome variables [33,34]. Such validity is particularly important for instruments designed for practical or diagnostic applications, for example, to identify organisational or individual factors that may affect employees’ occupational well-being, mental health, or intention to leave their job.

In this study, criterion (predictive) validity was tested using multiple linear regression analysis, with the three dimensions of the SRAQ-HP instrument as independent variables or predictors and the three theoretically significant indicators of occupational well-being as dependent variables or criteria: Model 1: Moral distress (MMD-HP) as dependent variable; Model 2: Professional burnout (CBI) as dependent variable; Model 3: Intent to leave job (ACT) as dependent variable.

To assess the criterion (predictive) validity of the SRAQ-HP instrument, a series of multiple linear regression analyses was conducted, in which the three SRAQ-HP dimensions (SRAQ1, SRAQ2, SRAQ3) were used as independent variables, while moral distress, burnout, and intention to leave were analysed as dependent variables. The regression results are summarised in Table 7.

When assessing criterion validity, several methodological and conceptual limitations must be acknowledged. First, the overall explained variance (R^2^) was relatively low in most of the models, which could indicate that the selected criterion variables and the SRAQ-HP dimensions are not conceptually linked closely enough. Second, statistical significance was achieved only in the first model, limiting the ability to draw generalisable conclusions. Third, the sample (n = 120), while sufficient for statistical analysis, may be structurally too homogeneous, which reduces the variability of the results. Fourth, only mean scale scores were used, which, while easily comparable, may not capture sufficient dimensional complexity. Fifth, the comparative instruments used (CBI, MMD-HP, ACT) have different conceptual structures and time orientations, which could affect the strength of the criterion relationships.

Summarising the results, the criterion validity of the SRAQ-HP instrument was partially supported, as most comparisons did not reach statistical significance and the amount of explained variance was low. The only model that provided strong evidence of criterion validity was the prediction of moral distress with the SRAQ1 subscale, showing a conceptually sound and statistically significant relationship between perceived resource adequacy and moral distress risks. This supports the theoretical assumption that insufficient staffing and limited resources in healthcare can act as triggers for moral distress when healthcare professionals experience a conflict between their professional values and the actual possibilities for providing adequate care.

Overall, the study results confirm that the SRAQ-HP is a conceptually coherent and statistically reliable instrument for assessing the adequacy of resources, support, and workload impact among healthcare professionals. Both EFA and CFA analyses confirmed a stable three-factor structure, while convergent and criterion validity analyses demonstrated the practical applicability of the instrument in empirical and policy contexts.

The SRAQ-HP can therefore be used as an effective tool to systematically identify weaknesses in human resource allocation and management and to support interventions aimed at improving the quality and sustainability of the healthcare work environment.

## 4. Discussion

The validation of the SRAQ-HP in Latvia revealed a three-factor structure, and this multidimensional configuration demonstrated both statistical robustness and theoretical coherence with the Job Demands–Resources (JD-R) model, which posits that adequate resources, organisational support, and manageable workload are key determinants of well-being and performance among healthcare professionals [7,8,9,10]. This structure is also consistent with models observed in similar instruments in international validation studies [35,36]. For example, hospital work environment assessments often identify factors corresponding to the dimensions of staff adequacy and management support. The Practice Environment Scale of the Nursing Work Index (PES-NWI) includes a subscale on Staff Adequacy and Resources, the items of which overlap conceptually with the first factor of the SRAQ-HP [35]. Interestingly, the international literature shows that the staff and resource adequacy factor tends to be very stable and replicable in studies from different countries [36]. For example, global studies on PES-NWI note that it is the staff adequacy and resources subscale that consistently emerges across all factor analysis solutions [13]. This is in line with our EFA/CFA results, which showed a clearly distinct staff adequacy factor. This finding aligns closely with the EFA/CFA results in the present study, where the first factor—adequacy of workload and staffing—demonstrated high loadings, strong internal consistency (α = 0.952), and high construct reliability (CR = 0.965), confirming its conceptual robustness.

Quality of care surveys also identify similar components. The Safety Attitudes Questionnaire (SAQ), which is widely used in healthcare, contains dimensions of Working Conditions and Perceptions of Management, which include statements on sufficient staffing and management support for staff [36]. SAQ studies have found that positive staff perceptions of staff adequacy correlate strongly with better patient safety outcomes. For example, in one study, nurses’ perception that there were enough staff more than doubled their assessment of patient safety [37]. This international finding is in line with the second SRAQ-HP factor we found, which highlights the impact of workload on the quality of care. In particular, when staff feel overburdened and under-resourced, this has a direct impact on the quality of care and safety. Similarly, studies in other countries have repeatedly documented the link between subjectively perceived staff shortages and increased burnout or errors in care [3]. This relationship was partially supported in the criterion validity results of the present study, where the SRAQ1 (resource adequacy) subscale significantly predicted moral distress, confirming that resource shortages can act as precursors of ethical strain and burnout in line with the JD-R and moral distress frameworks. These facts indicate that the dimensions constructed by the SRAQ-HP instrument are not isolated and reflect universal factors, as confirmed by comparable international studies.

The validation indicators for the SRAQ-HP instrument in Latvia also correspond to international experience. In our study, the overall internal consistency (Cronbach α) of the SRAQ-HP was high (above 0.94), similar to or even better than in analogous instruments elsewhere [35,36]. For example, the Cronbach α of the SAQ [36] subscales typically ranges between 0.70 and 0.90, and the PES-NWI [35] subscales often reach ~0.80 reliability, while the SRAQ-HP subscales exceeded this level. Furthermore, the convergent validity indicators (AVE ≥ 0.50 for two factors and CR > 0.85 for all) confirmed that the majority of the variance in observed items was explained by their latent constructs, except for one reverse-worded item (q14), which showed weaker loading and will be refined in future iterations. This shows that the SRAQ-HP items are very consistent in measuring their dimensions and that no topic ‘falls outside’ of the overall context.

The three-factor configuration—Adequacy of Workload and Staffing, Quality of Care, and Working Conditions and Support—demonstrated both empirical and theoretical convergence with international models of nursing work environments. This provides confidence that the factors identified are not context-specific but rather reflect globally recognised determinants of quality of care and staff well-being [38,39,40,41]. Moreover, the moderate inter-factor correlations observed in the CFA (r = 0.41–0.59) support the idea that these domains are distinct yet interdependent, consistent with the JD-R model’s assumption of interaction between job demands and resources. Therefore, the results of the SRAQ-HP can be meaningfully compared in an international context and used in broader multi-country studies, as the validation of the instrument meets high psychometric standards.

The practical implementation of the SRAQ-HP in Latvian healthcare opens up new opportunities for managing staff workload, improving the work environment, and optimising resource allocation. A validated instrument allows for a systematic measurement of healthcare staff’s perception of whether human resources are sufficient in a given workplace and setting. Such data are critically important at a time when Latvia and the world are experiencing staff shortages and overload in medicine [42]. This study’s results highlight that staff adequacy and support factors are central to maintaining both professional well-being and patient safety, emphasising the need for data-driven human resource management. With the help of the SRAQ-HP, hospital management can regularly ‘take the temperature’ of the workforce. Do staff feel overwhelmed or feel that there is a lack of support from colleagues and management, and where is this most reflected in the quality of care? Such feedback can guide targeted interventions to reduce moral distress and prevent burnout before they escalate.

Importantly, the SRAQ-HP also covers working conditions and support aspects such as management support. This means that the instrument can be used not only to assess ‘hard’ resources, the number of human resources, but also ‘soft’ factors such as organisational climate and management practices. If the SRAQ-HP shows a low support dimension in a hospital, this indicates a need to improve communication, engagement, and support mechanisms from management. Research shows that a positive, supportive management style can reduce the negative effects of stress even when resources are objectively scarce [38]. The SRAQ-HP therefore helps to identify opportunities for improvements in management behaviour, such as regular meetings with staff, a system of recognition for work performed, or clearer prioritisation processes. In practice, this contributes to an improved work environment, as employees who feel supported by management are more motivated and better able to cope with the workload [43]. These findings are consistent with the CFA results, which confirmed high loadings for the Working Conditions and Support factor (β range: 0.43–0.88), demonstrating that management and team climate are integral to perceived resource adequacy. This also has a direct impact on the quality of care because, as reminded by the second factor of the instrument, overload and an unsupportive environment can lower the level of patient care [44].

In the Latvian healthcare context, the SRAQ-HP fills a long-standing gap. Until now, there was no single, validated tool to measure aspects of staff resource adequacy from the perspective of employees themselves [11]. This became particularly relevant during the COVID-19 pandemic, when hospitals faced extreme overload and staff shortages. The present validation thus provides a methodological foundation for assessing perceived adequacy, complementing objective workload metrics such as NAS or TISS-28, and offering an integrated understanding of human resource strain. Using the SRAQ-HP, healthcare institutions can obtain evidence-based data on how staff perceive their work conditions, whether ICU staff feel sufficiently supported, or whether they face chronic understaffing. Such information is invaluable for promoting staff well-being and preventing burnout. By integrating these results with ongoing workforce monitoring, hospitals can implement early interventions and prevent the long-term consequences of moral distress and attrition.

It should be emphasised that the use of the SRAQ-HP is in line with Latvia’s health policy priorities. In recent years, official research and strategies have increasingly focused on psycho-emotional risks and staff well-being. The RSU study Working Conditions and Risks in Latvia 2019–2021 [38] concludes that psycho-emotional risks in the work environment (including lack of time and high workload) have become the most frequently cited problem in workplaces, including healthcare. Employers and employees most often cite lack of time, high workload, and the need to make quick, responsible decisions as the main risk factors [38]. This situation puts the emphasis on workload management and the psychological climate of the work environment. The authors of the study recommend more actively implementing psycho-emotional risk prevention measures and compiling examples of good practice, including the use of specialised digital tools in managing workplace risks [38]. The SRAQ-HP directly responds to this recommendation as a validated, evidence-based digital instrument capable of monitoring psycho-emotional strain and resource adequacy across healthcare settings [11]. Its use in hospitals is thus consistent with both institutional needs and national health policy directions aimed at improving working conditions and supporting staff mental health.

Given the validity and practical value of the instrument, the SRAQ-HP has a significant potential to be widely used in strategic and routine management processes in Latvian hospitals. First, the SRAQ-HP can be integrated as part of the quality management system of healthcare institutions. Similarly to how institutions regularly measure patient satisfaction or clinical indicators, they can introduce annual or semi-annual staff surveys with the SRAQ-HP to monitor trends in the work environment. The results of these surveys, collected anonymously, can be included in hospital management reports and discussed at board meetings, alongside financial and quality indicators. For example, a hospital may set a goal to improve staff resource adequacy by a defined margin over the next year and evaluate progress using repeated SRAQ-HP assessments. Such quantitative monitoring supports evidence-based decision-making and fosters organisational learning, turning subjective staff perceptions into actionable management indicators.

Second, the SRAQ-HP can serve as a tool for health policy planning more broadly. For example, the Ministry of Health or professional associations can use the instrument to carry out nationwide surveys on staff workload and resource adequacy. Such data can inform strategic decisions, such as the development of a health workforce development strategy. Improving the work environment and staff retention is already one of the three main directions of the already approved Health Workforce Development Strategy 2025–2029 [38]. The Strategy stresses the need to improve work environment conditions and employment conditions to create a motivating and engaging work environment for health specialists in the public sector [45,46]. The SRAQ-HP thus provides an empirical mechanism to monitor progress toward these strategic goals and to evaluate whether reforms translate into improved perceptions of staffing adequacy and management support.

Third, the SRAQ-HP can help healthcare organisations meet occupational safety and quality standards. EU and Latvian labour protection regulations require identification of psychosocial risks (e.g., excessive workload, stress, conflict) [47,48,49]. The SRAQ-HP can serve as a standardised methodology for conducting such assessments in healthcare. Regular staff surveys using the instrument provide documented evidence of psychosocial risk monitoring, supporting both quality accreditation and State Labour Inspectorate compliance. In addition, changes in SRAQ-HP results over time can be used as measurable indicators of intervention effectiveness—for example, improvement following workload redistribution or stress management programmes. This integration of psychometric data into the continuous quality improvement cycle represents an important advancement for Latvian healthcare institutions.

Finally, the potential of the SRAQ-HP is reflected in its ability to promote day-to-day improvement. Hospitals can anonymously compare unit-level results, share best practices, and identify strengths and weaknesses. Benchmarking across units and institutions creates opportunities for mutual learning and capacity building, enabling systemic improvement rather than isolated fixes. For example, if one unit shows higher scores for staff adequacy, its practices (e.g., flexible scheduling or supportive leadership) can be adopted elsewhere. On a national scale, if paediatric nurses consistently report lower resource adequacy than surgical nurses, this may prompt a policy review or targeted funding. In this way, the SRAQ-HP supports both micro-level (unit) and macro-level (policy) decision-making, helping to align staff well-being with patient safety and system sustainability.

In summary, the SRAQ-HP is not only a psychometrically validated research instrument but also a practical management tool with the potential to strengthen healthcare systems. It promotes a data-driven culture in healthcare, where staffing and well-being decisions are based on measurable evidence rather than assumptions. The instrument can become an integral component of change management, identifying critical areas, tracking progress, and reinforcing improvement through objective indicators. Ultimately, the use of the SRAQ-HP contributes to achieving national and international objectives, including the WHO’s call for improved working conditions and staff well-being [46]. In Latvia, where the health sector faces significant challenges related to staffing shortages and workforce ageing [50], the SRAQ-HP provides a timely, evidence-based solution for sustainable workforce management. It offers valuable feedback for enhancing both employee well-being and the quality of healthcare services, confirming the importance and necessity of implementing the instrument in Latvian hospitals [38,46].

Practical Interpretation, Study Limitations and Future Research:

While the SRAQ-HP demonstrates strong psychometric performance and broad practical applicability, several aspects require further clarification for use in everyday management and policy settings.

At present, no fixed cut-off points or normative benchmarks have yet been established for interpreting SRAQ-HP scores. However, average subscale values below 3.0 points (on a 5-point Likert scale) may tentatively indicate areas of concern or potential risk zones—such as insufficient staffing, reduced managerial support, or deteriorating perceptions of quality of care. In contrast, mean scores above 4.0 generally reflect a favourable work climate and adequate resource provision.

To ensure meaningful trend monitoring, it is recommended to administer the SRAQ-HP at least once or twice per year, ideally in connection with institutional quality and workforce assessments. These intervals allow the detection of gradual changes in perceived resource adequacy before they escalate into turnover, burnout, or patient-safety risks.

Future norm-referencing studies should establish empirically derived thresholds (e.g., percentile-based reference values) across professional groups and care settings to enable consistent benchmarking between hospitals and national systems.

Despite its strong results, this study has several limitations that should be considered.

First, the data were based on self-reported perceptions, which may be affected by response biases or social desirability.

Second, participation was voluntary and online, potentially limiting representativeness to more engaged professionals.

Third, the validation was conducted within a single country and time frame, which restricts generalisability and does not yet allow assessment of test–retest reliability or longitudinal stability.

Fourth, although most model-fit indices met acceptable thresholds, the RMSEA value (0.145 scaled) exceeded optimal limits, suggesting the need for further model refinement.

Fifth, item 14 showed a low factor loading, implying a possible conceptual ambiguity that warrants rewording or replacement in subsequent revisions.

Finally, although multiple criterion measures (CBI, MMD-HP, ACT) were used, their conceptual differences may have influenced the strength of observed correlations.

Future research should therefore focus on (a) cross-cultural and multi-institutional validation, (b) development of normative datasets and interpretation cut-offs, (c) evaluation of the SRAQ-HP’s sensitivity to change in intervention studies, and (d) examination of discriminant and predictive validity over time.

By addressing these aspects, future studies will help transform the SRAQ-HP from a validated measurement tool into a standardised benchmarking instrument for healthcare workforce management and organisational well-being assessment.

In practical terms, no fixed cut-off points have yet been established for interpreting SRAQ-HP results. However, mean subscale values below 3.0 on the five-point scale may tentatively indicate areas of concern—such as insufficient staffing, low perceived support, or declining quality of care—while average scores above 4.0 reflect favourable work environments. Until normative thresholds are developed, results should be interpreted contextually, comparing unit-level or institutional trends over time. Regular use of the SRAQ-HP, ideally once or twice per year, can help monitor workforce dynamics and identify early warning signs of overload or distress. Future research should include norm-referencing studies to define benchmark values and clinically meaningful change thresholds for different healthcare settings and professional groups.

## 5. Conclusions

This study conducted a comprehensive psychometric validation of the newly developed Staff Resource Adequacy Questionnaire for Healthcare Professionals (SRAQ-HP) among 1369 respondents representing diverse healthcare contexts in Latvia. The results confirmed a robust three-factor structure—adequacy of workload and staffing resources, quality of care, and working conditions/support—demonstrating theoretical and empirical coherence.

Although the confirmatory factor analysis (CFA) indicated an overall satisfactory model fit, the RMSEA value (0.145, scaled) exceeded conventional thresholds, suggesting that further model refinement may be needed. Other indices (CFI, TLI, SRMR) met or approached recommended standards, supporting the model’s adequacy for an initial validation phase.

High internal consistency (Cronbach’s α > 0.94) and acceptable convergent and criterion validity further confirmed the instrument’s reliability and construct soundness.

Nevertheless, the model fit limitations and the low loading of item 14 indicate areas that warrant further refinement. Future research should focus on re-evaluating or rewording item 14, testing the instrument’s stability over time (test–retest reliability), and assessing its applicability in different healthcare contexts and cultural settings.

Overall, the SRAQ-HP provides a psychometrically sound yet evolving tool for assessing perceptions of staff resource adequacy and organisational support in healthcare institutions, with strong potential for both research and practical implementation.

## Figures and Tables

**Figure 1 nursrep-15-00395-f001:**
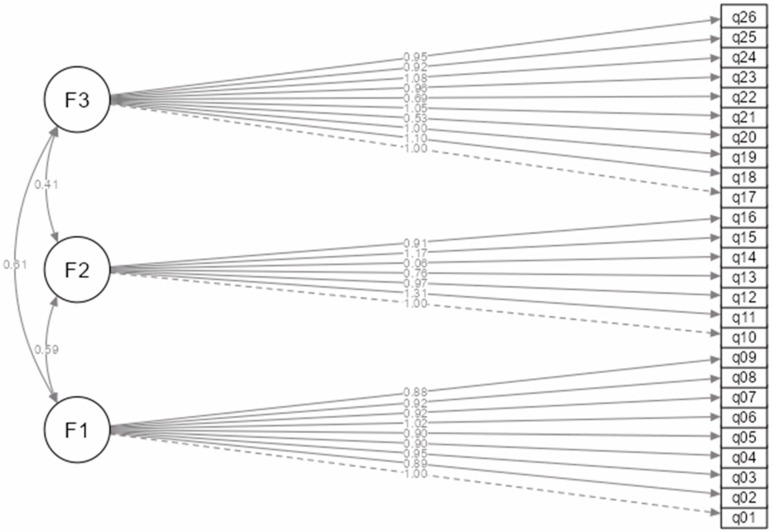
Path diagram for the three-factor CFA model (JASP output).

**Table 1 nursrep-15-00395-t001:** Results of the psychometric assessment.

Item	Mean	SD	Corrected Item-Total Correlation	Scale Mean if Item Deleted	Squared Multiple Correlation	Cronbach’s Alpha if Item Deleted	Skewness	Kurtosis	Kolmogorova-Smirnova Sig.	Shapiro-Wilka Sig.	Cronbach’s Alpha Based on Dimensions
**1 (1)**	2.49	1.29	0.791	70.72	0.897	0.948	0.466	−1.123	<0.001	<0.001	0.952
**2 (1)**	2.61	1.08	0.754	70.60	0.806	0.949	0.326	−0.960	<0.001	<0.001	
**3 (1)**	2.59	1.16	0.816	70.61	0.818	0.948	0.534	−0.777	<0.001	<0.001	
**4 (1)**	2.50	1.01	0.752	70.70	0.681	0.949	0.503	−0.462	<0.001	<0.001	
**5 (1)**	2.40	1.14	0.694	70.81	0.842	0.949	0.474	−0.884	<0.001	<0.001	
**6 (1)**	2.46	1.02	0.861	70.74	0.888	0.948	0.555	−0.444	<0.001	<0.001	
**7 (1)**	2.44	1.09	0.783	70.77	0.801	0.949	0.591	−0.508	<0.001	<0.001	
**8 (1)**	2.40	1.11	0.752	70.81	0.820	0.949	0.609	−0.328	<0.001	<0.001	
**9 (1)**	2.38	1.09	0.715	70.82	0.814	0.949	0.550	−0.416	<0.001	<0.001	
**10 (2)**	3.30	1.31	0.586	69.90	0.689	0.951	−0.361	−1.079	<0.001	<0.001	0.776
**11 (2)**	2.72	1.06	0.770	70.48	0.832	0.949	0.302	−0.887	<0.001	<0.001	
**12 (2)**	3.64	1.05	0.494	69.56	0.745	0.951	−1.038	0.630	<0.001	<0.001	
**13 (2)**	3.95	0.872	0.349	69.26	0.705	0.953	−0.968	1.070	<0.001	<0.001	
**14 (2)**	4.29	0.871	−0.001	68.92	0.489	0.955	−1.194	0.757	<0.001	<0.001	
**15 (2)**	2.34	0.967	0.662	70.86	0.800	0.950	0.583	0.048	<0.001	<0.001	
**16 (2)**	2.57	1.14	0.535	70.64	0.694	0.951	0.370	−0.760	<0.001	<0.001	
**17 (3)**	2.83	1.15	0.719	70.37	0.721	0.949	0.025	−1.270	<0.001	<0.001	0.903
**18 (3)**	2.33	1.15	0.754	70.88	0.789	0.949	0.524	−0.674	<0.001	<0.001	
**19 (3)**	2.79	1.17	0.670	70.42	0.634	0.950	0.220	−1.121	<0.001	<0.001	
**20 (3)**	3.09	1.10	0.348	70.11	0.513	0.953	−0.183	−0.869	<0.001	<0.001	
**21 (3)**	3.20	1.20	0.694	70.01	0.713	0.949	−0.385	−0.995	<0.001	<0.001	
**22 (3)**	3.31	1.14	0.442	69.89	0.525	0.952	−0.424	−0.848	<0.001	<0.001	
**23 (3)**	2.96	1.20	0.640	70.24	0.755	0.950	−0.294	−1.269	<0.001	<0.001	
**24 (3)**	2.50	1.06	0.739	70.70	0.780	0.949	0.366	−0.711	<0.001	<0.001	
**25 (3)**	2.82	1.06	0.611	70.39	0.641	0.950	−0.092	−0.952	<0.001	<0.001	
**26 (3)**	2.30	1.09	0.660	70.90	0.575	0.950	0.605	−0.301	<0.001	<0.001	

**Table 2 nursrep-15-00395-t002:** Pattern Matrix of the 26-Item SRAQ-HP (Principal Axis Factoring, Promax Rotation).

Item	Factor				h^2^	Uniqueness
	1	2	3	4		
**1 (1)**	0.818	0.169	0.119	0.339	0.826	0.174
**2 (1)**	0.767	0.319	0.003	0.137	0.709	0.291
**3 (1)**	0.747	0.277	0.107	0.381	0.790	0.210
**4 (1)**	0.713	0.341	0.087	0.137	0.652	0.348
**5 (1)**	0.845	0.265	0.001	−0.128	0.800	0.200
**6 (1)**	0.818	0.370	0.167	0.108	0.856	0.154
**7 (1)**	0.592	0.501	0.065	0.270	0.679	0.321
**8 (1)**	0.744	0.252	0.048	0.293	0.705	0.295
**9 (1)**	0.726	0.239	0.128	0.181	0.633	0.367
**10 (2)**	0.554	0.134	0.590	0.003	0.672	0.328
**11 (2)**	0.713	0.218	0.329	0.251	0.726	0.274
**12 (2)**	0.206	0.293	0.820	0.041	0.801	0.199
**13 (2)**	0.073	0.208	0.817	0.024	0.716	0.284
**14 (2)**	−0.111	−0.119	0.705	−0.032	0.525	0.475
**15 (2)**	0.822	0.199	−0.071	−0.005	0.720	0.280
**16 (2)**	0.491	0.562	−0.128	−0.257	0.639	0.361
**17 (3)**	0.346	0.536	0.239	0.457	0.673	0.327
**18 (3)**	0.333	0.768	0.102	0.265	0.780	0.220
**19 (3)**	0.291	0.586	0.250	0.334	0.602	0.398
**20 (3)**	0.122	0.083	0.061	0.770	0.618	0.382
**21 (3)**	0.450	0.449	−0.068	0.524	0.684	0.316
**22 (3)**	0.129	0.405	−0.164	0.632	0.607	0.393
**23 (3)**	0.235	0.763	0.212	0.088	0.689	0.311
**24 (3)**	0.422	0.702	0.058	0.175	0.705	0.295
**25 (3)**	0.208	0.746	0.050	0.204	0.643	0.357
**26 (3)**	0.450	0.501	0.230	0.064	0.511	0.489
**Eigenvalue after rotation**	7.956	4.940	2.667	2.390		
**% of variance**	30.599	19.000	10.258	9.193		

**Table 3 nursrep-15-00395-t003:** Pattern Matrix for the Three-Factor Solution (PAF with Promax Rotation, Fixed Number of Factors = 3).

Item	Factor			h^2^	Uniqueness
	1	2	3		
**1 (1)**	0.798	0.318	0.108	0.749	0.251
**2 (1)**	0.782	0.310	0.025	0.709	0.291
**3 (1)**	0.736	0.430	0.102	0.737	0.263
**4 (1)**	0.729	0.328	0.111	0.652	0.348
**5 (1)**	0.875	0.099	0.043	0.777	0.223
**6 (1)**	0.837	0.328	0.195	0.845	0.155
**7 (1)**	0.616	0.540	0.091	0.678	0.322
**8 (1)**	0.738	0.357	0.049	0.675	0.325
**9 (1)**	0.727	0.275	0.138	0.623	0.377
**10 (2)**	0.549	0.080	0.600	0.668	0.332
**11 (2)**	0.701	0.300	0.329	0.689	0.311
**12 (2)**	0.212	0.234	0.838	0.801	0.199
**13 (2)**	0.073	0.162	0.827	0.716	0.284
**14 (2)**	−0.137	−0.119	0.690	0.509	0.491
**15 (2)**	0.837	0.128	−0.046	0.719	0.281
**16 (2)**	0.569	0.260	−0.048	0.393	0.607
**17 (3)**	0.356	0.690	0.247	0.664	0.336
**18 (3)**	0.386	0.750	0.151	0.735	0.265
**19 (3)**	0.316	0.652	0.275	0.601	0.399
**20 (3)**	0.066	0.546	−0.002	0.302	0.698
**21 (3)**	0.452	0.666	−0.070	0.653	0.347
**22 (3)**	0.123	0.711	−0.181	0.554	0.446
**23 (3)**	0.300	0.636	0.275	0.570	0.430
**24 (3)**	0.476	0.641	0.110	0.649	0.351
**25 (3)**	0.267	0.699	0.103	0.570	0.430
**26 (3)**	0.487	0.412	0.272	0.481	0.519
**Eigenvalue after rotation**	5.563	5.563	5.563		
**% of variance**	32.111	21.397	10.805		

**Table 4 nursrep-15-00395-t004:** Comparative goodness-of-fit indicators of the CFA model.

	Model	Scaled
Comparative Fit Index (CFI)	0.968	0.898
Tucker–Lewis Index (TLI)	0.964	0.888
Bentler-Bonett Non-normed Fit Index (NNFI)	0.964	0.888
Relative-Noncentrality Index (RNI)	0.968	0.898
Bentler-Bonett Normed Fit Index (NFI)	0.967	0.894
Bollen’s Relative Fit Index (RFI)	0.964	0.884
Bollen’s Incremental Fit Index (IFI)	0.968	0.898
Parsimony Normed Fit Index (PNFI)	0.881	0.815

**Table 5 nursrep-15-00395-t005:** Absolute goodness-of-fit indices (SRMR and RMSEA with 90% CI).

				90% Confidence Intervals		
Type	SRMR	RMSEA	Lower	Upper	RMSEA *p*	χ^2^/df
Conventional	0.114	0.178	0.175	0.181	<0.001	44.4
Robust	0.100					
Scaled	0.100	0.145	0.142	0.148	<0.001	42.7

**Table 6 nursrep-15-00395-t006:** Factor loadings and contribution of observed variables (standardised regression loadings and 95% confidence intervals).

95% Confidence Intervals								
Latent	Observed	Estimate	SE	Lower	Upper	β	z	*p*
**Factor 1**	1	1.0000	0.00000	1.00000	1.000	0.9327		<0.001
	2	0.8924	0.01032	0.87213	0.913	0.8323	86.51	<0.001
	3	0.9476	0.00684	0.93423	0.961	0.8839	138.45	<0.001
	4	0.9024	0.00989	0.88301	0.922	0.8417	91.29	<0.001
	5	0.9017	0.00807	0.88584	0.917	0.8410	111.71	<0.001
	6	1.0185	0.00695	1.00487	1.032	0.9500	146.63	<0.001
	7	0.9227	0.00990	0.90330	0.942	0.8606	93.22	<0.001
	8	0.9191	0.00754	0.90435	0.934	0.8573	121.88	<0.001
	9	0.8772	0.00992	0.85775	0.897	0.8182	88.45	<0.001
**Factor 2**	10	1.0000	0.00000	1.00000	1.00	0.7322		<0.001
	11	1.4406	0.03931	1.36351	1.518	0.9560	36.34	<0.001
	12	1.0663	0.3545	0.99681	1.136	0.7076	30.08	<0.001
	13	0.8352	0.03753	0.76167	0.909	0.5543	22.25	<0.001
	14	0.0676	0.03950	−0.0098	0.145	0.0448	1.71	0.087
	15	1.2955	0.03436	1.22813	1.363	0.8597	37.70	<0.001
	16	1.1034	0.03630	1.03222	1.175	0.6636	30.39	<0.001
**Factor 3**	17	1.0000	0.00000	1.00000	1.000	0.8027		<0.001
	18	1.0952	0.01265	1.07045	1.120	0.8792	86.61	<0.001
	19	0.9979	0.01785	0.96291	1.033	0.8011	55.90	<0.001
	20	0.5337	0.02114	0.49224	0.575	0.4284	25.25	<0.001
	21	1.0551	0.01806	1.01972	1.091	0.8470	58.43	<0.001
	22	0.6941	0.02144	0.65204	0.736	0.5572	32.37	<0.001
	23	0.9572	0.1650	0.92489	0.990	0.7684	58.01	<0.001
	24	1.0798	0.01453	1.05133	1.108	0.8668	74.32	<0.001
	25	0.9223	0.01753	0.88797	0.957	0.7404	52.62	<0.001
	26	0.9472	0.01900	0.90995	0.984	0.7603	49.86	<0.001

**Table 7 nursrep-15-00395-t007:** Multiple linear regression predicting moral distress (MMD-HP).

Predictor	β (MMD-HP)	*p*	β (CBI)	*p*	β (ACT)	*p*
SRAQ1	0.420	0.007	−0.126	0.300	−0.164	0.240
SRAQ2	−0.202	0.125	−0.091	0.479	0.189	0.148
SRAQ3	−0.278	0.039	0.121	0.323	−0.089	0.512
R^2^/Adj.R^2^	0.21/0.18		0.08/0.04		0.06/0.03	

## Data Availability

The datasets produced and examined in this study can be obtained from the corresponding author upon a reasonable request. All data generated or analysed during this study are provided within the published article. The data utilised in this study is confidential.

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
