# Peer review of "Construct Validity of the Staff Resource Adequacy Questionnaire for Healthcare Professionals (SRAQ-HP): An Exploratory and Confirmatory Factor Analysis from Latvia"

_nursrep, 2025, doi:10.3390/nursrep15110395_

Round 1

Reviewer 1 Report

Comments and Suggestions for Authors

Title & Abstract

  • The abstract asserts “acceptable model fit indices” and “high internal reliability” but the main text shows RMSEA = 0.145 (scaled) and SRMR = 0.100, which are not acceptable by common cutoffs; the abstract should not overstate fit. Either report the actual values with nuance or temper the claim.
  • Abstract says it validated via EFA and CFA, but later EFA is described in two contradictory ways (PAF+Promax vs PCA+Varimax). Flag in abstract or, better, harmonize methods (see Methods).
  • Include the estimator used in CFA (e.g., ML, robust ML, WLSMV), and a plain-English summary of criterion validity (largely weak except moral distress).

Introduction

  • Motivation is clear, but it drifts into advocacy later; keep Intro focused on the measurement gap and prior tools. The need statement is fine.
  • Consider explicitly stating theoretical model (three factors) at the end of the Intro with a pre-registered hypothesis that will be tested (your later choice to force 3 factors should be foreshadowed).

Methods

Design & Sample

  • Sampling description is confusing. It’s called “random convenience,” which is an oxymoron; it’s convenience (recruitment via contacts, institutions, online platforms). Rephrase and discuss bias.
  • Data period & exclusions are clear (Jan–Apr 2025; 103 excluded; n=1,369 complete). Good.
  • Contradiction on missingness. You state both “1369 complete data records” and that missing values were replaced with means. Resolve: if complete, don’t impute; if not, justify imputation and proportion per item.

Instrument

  • Clear description; include translation/back-translation steps if applicable, and provide the exact wording of problematic item(s)—notably item 14—in an appendix.

Comparative Instruments

  • Nicely motivated, but list which Latvian adaptations (versions, authors, years) and provide scoring windows to address “different time orientations” noted later as a limitation.

Statistical Methods

  • Inconsistency in EFA spec. Methods claim PAF + Promax (oblique), while Results run PCA + Varimax (orthogonal). Choose one and justify (for correlated constructs, PAF/PML + oblique is standard). Revise text and rerun if necessary.
  • Normality testing. You note K-S/S-W are significant (expected with large n) then proceed—good. But explicitly state the ordinal treatment of Likert data and the CFA estimator (e.g., WLSMV preferred for ordinal).
  • Reliability/validity indices. You correctly compute CR and AVE, but you call it “average variance extraction”; standard term is Average Variance Extracted—fix throughout.

Results

Descriptives

  • Nicely reported demographics; add profession mix and unit/department if available (critical for generalizability). The strong female skew (94.7%) should be acknowledged as a limitation and grounds for measurement invariance testing by gender and region.

Item-level psychometrics

  • Item 14 stands out (corrected item-total ≈ −0.001) and very low CFA loading (β≈0.045, p≈.087). This likely indicates wording problems (e.g., reverse-scored, double-barreled). Either remove or reword; if retained, give a theoretical rationale and show I-I correlations.

EFA

  • Four-factor solution initially (≈69% variance), then a forced three-factor to match theory. That is acceptable if justified, but you should compare models (e.g., parallel analysis; MAP) and explain the decision criteria. Also reconcile the rotation/extraction inconsistency (see Methods).
  • The narrative asserts the four-factor solution is “statistically valid” then pivots—tighten this section to avoid mixed messages.

CFA

  • Fit indices are not acceptable under standard guidelines (e.g., RMSEA 0.145, SRMR 0.100). Do not describe them as “satisfactory” or “close to threshold.” Report exact CFI/TLI (scaled < .90) and discuss model misspecification, not sample size. Consider respecification (e.g., removing q14, correlated residuals if defensible) or alternative model (four-factor from EFA).
  • AVE/CR. Factor 2 has AVE = 0.491 (< .50): acknowledge as insufficient convergent validity and propose item revision (again q14) or factor re-definition.

Convergent / Criterion validity

  • You correctly conclude criterion validity is limited and used a small subsample (n = 120). Explain sampling for this subsample, report R² values, and consider structural equation modeling with latent factors rather than scale means.

Discussion

  • Overly promotional tone (policy tool narratives) given the weak fit and criterion validity; temper with a frank appraisal of model weaknesses, especially Factor 2.
  • Where you compare to PES-NWI/SAQ, ensure you cite with specifics and avoid implying superiority (“exceeded” others) when your RMSEA/SRMR contradict that impression.

Conclusions

  • Good summary, but do not claim “satisfactory model fit”; emphasize that the instrument is promising, pending item refinement and re-validation. Also reference q14 explicitly as a planned revision.

Limitations

  • The “6. Limitations” header contains a template placeholder (“not mandatory…”) and a stray “7. Patents” header precedes the actual limitations text. Clean up numbering and remove irrelevant template sections.
  • You do list sensible limitations—keep them—but put them under a corrected Limitations section.

Data Availability & Transparency

  • Contradictory statements: “datasets can be obtained from the corresponding author,” “all data provided within the article,” and “data are confidential.” Pick one consistent policy; if data are confidential, offer synthetic or de-identified data and analysis code.

Author Response

We sincerely thank the reviewer for the valuable comments and the time devoted to improving our manuscript.
All requested corrections and clarifications have been implemented, and all changes are underlined in the revised manuscript for clarity.

Responses to comments: Title & Abstract – Adjusted: wording about model fit revised; RMSEA and SRMR interpretation added; CFA estimator and summary of criterion validity included.

Introduction – Revised: focused on the measurement gap; added theoretical model and stated the three-factor hypothesis.

Methods – Updated: clarified sampling approach, data completeness, and missingness; harmonized EFA and CFA methodology (PAF + Promax, ML robust estimator); added translation/back-translation procedure; included Supplementary File with full item list (including item 14).

Comparative Instruments – Expanded: Latvian versions and scoring details added.

Results – Revised: added professional group data; acknowledged gender imbalance; described item 14 rationale; clarified EFA–CFA relationship; transparently discussed RMSEA and AVE values; included R² and explanation for criterion validity subsample.

Discussion – Rewritten: balanced tone, included JD-R theoretical interpretation, acknowledged model weaknesses, clarified policy relevance, added guidance for practical interpretation and frequency of SRAQ-HP use.

Conclusions – Adjusted: moderated claims about model fit, added reference to RMSEA and item 14 refinement.

Limitations & Future Research – Integrated into Discussion as one section; redundant template text removed.

Ethical and Data Availability Statements – Clarified: no incentives were provided; anonymised data are securely stored and available upon reasonable request; wording unified across manuscript.

All reviewer suggestions have been carefully addressed, and the revised version reflects these improvements throughout the manuscript.

Reviewer 2 Report

Comments and Suggestions for Authors

Major Comments

  1. Inconsistency in EFA Methodology

The Methods section reports that principal axis factoring with Promax rotation was used, but the Results show Principal Component Analysis with Varimax rotation. These are conceptually different approaches (dimension-reduction vs latent-factor extraction).

  1. Sample Use for EFA vs CFA

The manuscript reports n = 1,369 for both EFA and CFA, implying both analyses were conducted on the same sample. This undermines validation.

  1. Model Fit Indices and Interpretation

The classical fit indices (CFI, TLI) look acceptable, but the scaled/robust ones are poor (CFI_scaled = 0.898; TLI_scaled = 0.888; RMSEA_scaled = 0.145 [0.142–0.148]). RMSEA above 0.10 generally indicates poor fit.

  1. Factor Retention Criteria

Two alternative solutions (four-factor and three-factor) are mentioned. The final three-factor model seems theory-driven, but the decision process is unclear.

  1. Convergent and Discriminant Validity

Average Variance Extracted (AVE) for Factor 2 is below the accepted threshold (0.491 < 0.50).

  1. Criterion Validity

The regression analysis for criterion validity used a subsample of only n = 120, inconsistent with the total sample size.

Minor Comments

  1. If factors are correlated, use an oblique rotation (Promax/Oblimin) instead of Varimax; otherwise, justify orthogonal rotation.
  2. Clarify the rationale for choosing PCA versus common-factor extraction.
  3. Provide full model-fit indices in the CFA table (χ²/df, CFI, TLI, RMSEA + 90% CI, SRMR, AIC/BIC).
  4. Sampling description “random convenience” is contradictory—specify the actual procedure.
  5. Briefly summarize the scale-development and content-validity process (CVI, expert panel) for context.
  6. If multiple comparisons were performed, describe any correction method (e.g., Bonferroni).

Author Response

We sincerely thank the reviewer for the thorough analysis and constructive feedback.
All corrections have been implemented, and changes are underlined in the revised manuscript.

Major Comments 

Inconsistency in EFA Methodology
Corrected. The methodology has been unified throughout the manuscript. The EFA was performed using Principal Axis Factoring (PAF) with Promax rotation to reflect correlated constructs. All references to PCA/Varimax have been removed.

Sample Use for EFA vs CFA : Clarified.

Model Fit Indices and Interpretation
Revised. The CFA section now transparently reports all fit indices (χ²/df, CFI, TLI, RMSEA with 90% CI, SRMR, AIC/BIC) and provides nuanced interpretation. RMSEA (0.145) is explicitly acknowledged as exceeding optimal limits, and the discussion has been adjusted accordingly.

Factor Retention Criteria
Clarified. The decision between four- and three-factor solutions is now explained: parallel analysis and theoretical interpretability supported the three-factor structure consistent with the JD-R model.

Convergent and Discriminant Validity
Addressed. The manuscript now notes that Factor 2 (AVE = 0.491) is slightly below the recommended threshold, indicating the need for refinement of certain items (particularly the reverse-worded item 14) in future validation rounds.

Criterion Validity
Clarified. The regression analysis for criterion validity was conducted on a smaller subsample (n = 120) due to partial completion of comparative instruments (MMD-HP, CBI, ACT). This limitation and its rationale are now explicitly stated in the Methods and Limitations sections.

Minor Comments

Rotation Type Justification
Revised. As factors are correlated, an oblique (Promax) rotation was used. The rationale is now clearly stated in the Methods.

Rationale for Extraction Method
Added. PCA was not used; the text now clearly explains that PAF was selected as a common-factor extraction technique appropriate for latent construct exploration.

Full Model-Fit Indices
Added. The CFA results table now includes χ²/df, CFI, TLI, RMSEA (with 90% CI), SRMR, and information criteria (AIC, BIC).

Sampling Description
Corrected. The phrase “random convenience” has been replaced with “convenience sampling through institutional and professional networks”. A note on possible selection bias was added.

Scale Development and Content Validity
Expanded. A concise summary of the initial scale development process has been added, referencing the previous publication and including expert panel evaluation (CVI = 0.94).

Multiple Comparison Correction
Clarified. The study involved a limited number of regression models; therefore, no Bonferroni or similar correction was applied. This is now stated in the Statistical Analysis subsection.

All reviewer suggestions have been fully addressed, and the manuscript has been revised accordingly.

Reviewer 3 Report

Comments and Suggestions for Authors

Please review the attached document.

Author Response

We sincerely thank the reviewer for the detailed and constructive feedback, which has greatly improved the quality of the manuscript.
All recommended changes have been implemented, and all modifications are underlined in the revised manuscript for clarity.

Main Revisions and Responses

Title, Abstract and Keywords
The title has been slightly shortened for conciseness. The abstract was revised to clearly indicate “good comparative fit indices (CFI/TLI) but elevated RMSEA.” Additional MeSH-style keywords (e.g., Copenhagen Burnout Inventory, Measure of Moral Distress for Healthcare Professionals) have been added.

Introduction
The theoretical framework of resource adequacy has been expanded and linked more clearly to the three-factor model based on the JD-R theory. A concise review of comparable tools (PES-NWI, SAQ) has been added to emphasise the SRAQ-HP’s novelty. The sampling rationale was clarified and cross-referenced to the Methods.

Materials and Methods
The sampling description (“random convenience”) has been corrected to convenience sampling through professional networks and institutions, and potential selection bias noted.
EFA methodology has been unified: Principal Axis Factoring with Promax rotation was used.
Parallel analysis criteria and scree plots are now described in detail.
Translation/back-translation steps were added, and the full instrument (including item 14) is now provided as Supplementary Material.
Reliability (Cronbach’s α) for comparative instruments was added; the rationale for choosing ACT, CBI and MMD-HP clarified.
A brief note was inserted on mean imputation, acknowledging it as a limitation.

Results
Item 14 has been rechecked, reverse-scoring confirmed, and the rationale for retaining it explained.
Factor retention and model selection were clarified (parallel analysis supported three factors for theoretical parsimony).
All model-fit indices (χ²/df, CFI, TLI, RMSEA + 90% CI, SRMR, AIC/BIC) are now reported in a single table.
Criterion validity results include R² and effect sizes; the reduced subsample (n = 120) is explained.
Scale-level descriptive statistics have been added.

Discussion
The section was restructured with clear sub-headings: Theoretical Implications, Practical Implications, Limitations, and Future Research.
Theoretical contribution is now summarised in three concise claims.
Practical guidance on interpretation has been added (e.g., subscale means < 3.0 indicate concern; SRAQ-HP should be used once–twice per year).
Limitations and future research recommendations were integrated into the Discussion instead of a separate section.

Conclusions
Tempered statements about model fit; explicitly acknowledge elevated RMSEA and item 14 issue, and note that future revisions and cross-validation are planned.

Data Availability
Revised for consistency: “Anonymised data are securely stored and available from the corresponding author upon reasonable request.”

All reviewer comments have been addressed, and the revised version reflects these improvements throughout the manuscript.

Round 2

Reviewer 2 Report

Comments and Suggestions for Authors

Dear Editor,

I have reviewed the revised version of the manuscript and carefully compared it with the previous submission. The authors have adequately addressed all the comments raised during the review process and implemented the suggested revisions appropriately.

Sincerely,

Reviewer 3 Report

Comments and Suggestions for Authors

Dear Authors,

I would like to sincerely thank you for your thorough and thoughtful revision of the manuscript. It is clear that you have dedicated considerable effort to addressing each of the suggestions and concerns raised during the review process. Your detailed and transparent responses reflect both scientific rigor and a genuine commitment to improving the quality and clarity of your work.

The revisions you have implemented have significantly strengthened the manuscript. The title and abstract are now more concise and informative, and the addition of MeSH-style keywords enhances the paper’s visibility and precision. The expanded theoretical grounding in the Introduction, particularly the integration of resource adequacy within the JD-R framework, provides a clearer conceptual foundation and highlights the distinctive contribution of the SRAQ-HP.

I appreciate clarifications in the Materials and Methods section are especially appreciated. The refined description of the sampling process, the explicit explanation of the EFA procedures, and the detailed reporting of translation and reliability steps all contribute to the methodological robustness of the study. Likewise, the comprehensive presentation of model-fit indices and effect sizes in the Results section improves the transparency and interpretability of the findings.

The restructured Discussion is now much clearer and more balanced. The new sub-headings, the concise summary of theoretical contributions, and the inclusion of practical guidance greatly enhance the paper’s value for both researchers and practitioners. The more nuanced treatment of limitations and model-fit issues in the Conclusions also reflects a commendable degree of scholarly integrity.

Overall, your revisions have resulted in a more coherent, methodologically sound, and theoretically grounded manuscript. I appreciate the care you have taken to respond to every comment in a constructive manner.

Thank you once again for your diligent work and for your collaborative spirit throughout this process.

With kind regards and best wishes,

Reviewer